



# A Novel Framework for Molecular Characterization of Atmospheric Organic Aerosol Based on Collision Cross Section and Mass-to-Charge Ratio

X. Zhang[1], J. E. Krechmer[2,3], M. Groessl[4], W. Xu[1], S. Graf[4], M. Cubison[4],

J. T. Jayne[1], J.L. Jimenez[2,3], D. R. Worsnop[1], and M. R. Canagaratna[1]

[1] Center for Aerosol and Cloud Chemistry, Aerodyne Research Inc., Billerica, MA 01821, USA

[2] Department of Chemistry and Biochemistry, University of Colorado, Boulder, CO 80309, USA

[3] Cooperative Institute for Research in Environmental Sciences, Boulder, CO 80309, USA

[4] TOFWERK, CH-3600 Thun, Switzerland

*Correspondence to*: M. R. Canagaratna (mrcana@aerodyne.com)



**Abstract**
A new metric is introduced for representing the molecular signature of atmospheric
organic aerosols, the collision cross section ($\Omega$), a quantity that is related to the structure
and geometry of molecules and is derived from ion mobility measurements. By
combination with the mass-to-charge ratio ($m/z$), a two-dimensional $\Omega - m/z$ space is
developed to facilitate the comprehensive investigation of the complex organic aerosol
mixture. A unique distribution pattern of chemical classes, characterized by functional
groups including amine, alcohol, carbonyl, carboxylic acid, ester, and organic sulfate, is
developed on the 2-D $\Omega - m/z$ space. Species of the same chemical class, despite
variations in the molecular structures, tend to situate as a narrow band on the space and
follow a trend line. Reactions involving changes in functionalization and fragmentation
can be represented by the directionalities along or across these trend lines, thus allowing
for the interpretation of mechanisms associated with the formation and evolution of
atmospheric organic aerosol. The characteristics of trend lines for a variety of
functionalities that are commonly present in ambient aerosols can be predicted by the
core model simulations, which provide a useful tool to identify the chemical class to
which an unknown species belongs on the $\Omega - m/z$ space. Within the band produced by
each chemical class on the space, molecular structural assignment can be achieved by
utilizing collision induced dissociation as well as by comparing the measured collision
cross sections in the context of those obtained via molecular dynamics simulations.





## 1. Introduction

Organic aerosol (OA) constitutes a major fraction of sub-micrometer atmospheric particulate matter and comprises a complex and dynamic system linking source emission, chemical transformation, and phase partitioning (Hallquist et al., 2009). It consists of a multitude of organic species that arise from primary emissions and secondary productions. Once in the atmosphere, OA species actively evolve via gas-particle conversion and multiphase chemistry. The complexity and dynamic behaviors of ambient OA have rendered identification of major pathways contributing to OA budget difficult and have limited our capability to evaluate its impact on human health and global climate.

Several two-dimensional frameworks have been developed in an effort to deconvolve the complexity of OA mixtures and visualize their atmospheric transformations. The Van Krevelen diagram, which scatter plots the hydrogen-to-carbon atomic ratio (H:C) and the oxygen-to-carbon atomic ratio (O:C), has been widely used to represent the bulk elemental composition and the degree of oxygenation of organic aerosol (Heald et al., 2010). The average carbon oxidation state ($\overline{OS}_C$), a quantity that necessarily increases upon oxidation, can be estimated from the elemental ratios (Kroll et al., 2011). When coupled with carbon number ($n_C$), it provides constraints on the chemical composition of organic aerosol and defines key classes of atmospheric processes based on the unique trajectory of the evolving OA composition on the $\overline{OS}_C - n_C$ space. The degree of oxidation has also been combined with the volatility (expressed as the effective saturation concentration, $C^*$), forming a 2-D volatility basis set to describe the coupled aging and phase partitioning of organic aerosol (Donahue et al., 2012). These three spaces are designed to represent fundamental properties of the OA mixture and provide insight into the OA chemical evolution in the atmosphere. Organic aerosol components span large varieties in the physicochemical properties. Species of similar volatility or elemental composition can differ vastly in structures and functionalities. One weakness of these frameworks is that they do not provide information on the OA components at molecular level.



In this article we introduce a new framework that is based on the collision cross
section ($\Omega$), a quantity that is related to the structure and geometry of a molecule. The
collision cross section of a charged molecule determines its mobility as it travels through
a neutral buffer gas such as $N_2$ under the influence of a weak and uniform electric field.
Species with open conformation undergo more collisions with buffer gas molecules and
hence travel more slowly than the compact ones (Shvartsburg et al., 2000; Eiceman et al.,
2013). Mobility measurements are usually performed with an Ion Mobility Spectrometer
(IMS), where ions are separated mainly on the basis of their size, geometry, as well as
interactions with the buffer gas. The combination of IMS with a Mass Spectrometer (MS)
allows for further selection of ions based on their mass-to-charge ratios. The resulting
IMS-MS plot provides separation of molecules according to two different properties:
geometry (as reflected by the collision cross section) and mass (as reflected by the mass-
to-charge ratio) (Kanu et al., 2008). The Ion Mobility Spectrometry - Mass Spectrometry
(IMS-MS) analytical technique has been widely employed in the fields of biochemistry
(McLean et al., 2005; Liu et al., 2007; Dwivedi et al., 2008; Roscioli et al., 2013; Groessl
et al., 2015) and homeland security (Eiceman and Stone, 2004; Ewing et al., 2001;
Fernandez-Maestre et al., 2010). To our knowledge, the application of IMS-MS to study
organic species in the atmosphere, however, has only been explored very recently
(Krechmer et al., 2016).
We propose a two-dimensional collision cross section vs. mass-to-charge ratio ($\Omega -$
$m/z$) space to facilitate the comprehensive investigation of complex OA mixtures. Despite
the typical complexity of the detailed molecular mechanism of OA production and
evolution, oxidized molecules that constitute OA can be characterized by their distinctive
functional groups (Zhang and Seinfeld, 2013). We show that the investigated organic
classes ($m/z < 600$), characterized by functional groups including amine, alcohol,
carbonyl, carboxylic acid, ester, and organic sulfate, exhibit unique distribution patterns
on the $\Omega - m/z$ space. Species of the same chemical class, despite variations in the
molecular structures, tend to develop a narrow band and follow a trend line on the space.
Reactions involving changes in functionalization and fragmentation can be represented
by directionalities along or across these trend lines. The locations and slopes of the
measured trend lines are shown to be predicted by the core model (Mason et al., 1972),



which characterizes the ion-neutral interactions as elastic sphere collisions. Within the
narrow band produced by each chemical class on the $\Omega - m/z$ space, molecular structural
assignment is achieved with the assistance of collision induced dissociation analysis.
Measured collision cross sections are also shown to be consistent with theoretically
predicted values from the trajectory method (Mesleh et al., 1996; Shvartsburg and
Jarrold, 1996) and are used to identify isomers that are separated from an isomeric
mixture.

## 2. Collision Cross Section Measurements

2.1 Materials
A collection of chemical standards (ACS grade, $\geq$ 96%, purchased from Sigma
Aldrich, St. Louis, MO, USA), classified as amines, alcohols, carbonyls, carboxylic
acids, esters, phenols, and organic sulfates, were used to characterize the performance of
IMS-MS. These chemicals were dissolved in an HPLC-grade solvent consisting of a 70%
methanol / 29% water with 1% formic acid, at a concentration of approximately 10 µM.
2.2 Instrumentation
Ion mobility measurements were performed using an Electrospray Ionization (ESI)
Drift-Tube Ion Mobility Spectrometer (DT-IMS) interfaced to a Time-of-Flight Mass
Spectrometer (TOFMS). The instrument was designed and manufactured by TOFWERK
(Switzerland), with detailed descriptions and schematics provided by several recent
studies (Kaplan et al., 2010; Zhang et al., 2014; Groessl et al., 2015; Krechmer et al.,
2016). In the next few paragraphs, we will present the operating conditions of the ESI-
IMS-TOFMS instrument.
Solutions of chemical standards were delivered to the ESI source via a 250 µL gas-
tight syringe (Hamilton, Reno, NV, USA) held on a syringe pump (Harvard Apparatus,
Holliston, MA, USA) at a flow rate of 1 µL min$^{-1}$. A deactivated fused silica capillary
(360 µm OD, 50 µm ID, 50 cm length, New Objective, Woburn, MA, USA) was used as
the sample transfer line. The ESI source was equipped with an uncoated SilicaTip Emitter
(360 µm OD, 50 µm ID, 30 µm tip ID, New Objective, Woburn, MA, USA) and



connected to the capillary through a conductive micro union (IDEX Health & Science,
Oak Harbor, WA, USA). The ESI emitter was operated at both positive and negative
mode at a capillary voltage of $\pm$ (1.5 − 2.0) kV. The charged droplets generated at the
emitter tip migrate through a desolvation region in nitrogen atmosphere at room
temperature, where ions evaporate from the droplets and are introduced into the drift tube
through a Bradbury-Nielsen ion gate located at the entrance. The ion gate was operated in
the Hadamard Transform mode, with a closure voltage of $\pm$ 50 V and an average gate
pulse frequency of $1.2 \times 10^3$ Hz. The drift tube was held at a constant temperature (340±3
K) and atmospheric pressure ($\sim$ 1019 *mbar*). A counter flow of $N_2$ drift gas was
introduced at the end of the drift region at a flow rate of 1.2 L min$^{-1}$. Ion mobility
separation was carried out at a typical filed strength of 300 − 400 V cm$^{-1}$, resulting in a
reduced electric field of approximately 1.4 − 1.8 Td. After exiting from the drift tube,
ions were focused into TOFMS through a pressure-vacuum interface that includes two
segmented quadrupoles that were operated at $\sim$ 2 *mbar* and $\sim 5 \times 10^{-3}$ *mbar*, respectively.
Collision Induced Dissociation (CID) of parent ions is achieved by adjusting the voltages
on the ion optical elements between the two quadruple stages (Kaplan et al., 2010).
The ESI-IMS-TOFMS instrument was operated in the *m/z* range of 40 to 1500 with a
total recording time of 90 s for each dataset. The Mass Spectrometer was calibrated using
a mixture of quaternary ammonium salts, reserpine, and a mixture of fluorinated
phosphazines (Ultramark 1621) in the positive mode and ammonium phosphate, sodium
dodecyl sulfate, sodium taurocholate hydrate, and Ultramark 1621 in the negative mode.
The ion mobility measurements were calibrated using tetraethyl ammonium chloride as
the instrument standard and 2,4-lutidine as the mobility standard, as defined shortly
(Fernández-Maestre et al., 2010). Mass spectra and ion mobility spectra were recorded
using the acquisition package "Acquility" (v2.1.0, http://www.tofwerk.com/acquility).
Post-processing was performed using the data analysis package "Tofware" (version 2.5.3,
www.tofwerk.com/tofware) running in the Igor Pro (Wavemetrics, OR, USA)
environment.


2.3 Calculations
The average velocity of an ion in the drift tube ($v_d$) is proportional to its characteristic
mobility constant ($K$ / cm$^2$ V$^{-1}$ s$^{-1}$) and the electric field intensity ($E_d$), provided that the
field is weak (McDaniel and Mason, 1973):

$$v_d = K\,E_d \tag{1}$$

Experimentally, ion mobility constants can be approximated from the time of ion clouds
spent in the drift tube ($t_d$ / s), given by the rearranged form of Equation (1):

$$t_d = \frac{1}{K}\,\frac{L_d^2}{V_d} \tag{2}$$

where $L_d$ (cm) is the length of the drift tube and $V_d$ (V) is the drift voltage. In the present
study, drift time measurements were carried out at six different drift voltages ranging
from 5 kV to 8 kV in ~ 1019 $mbar$ of nitrogen gas at 340 K (Figure S1 in the
supplement). The ion mobility constant ($K$) is derived by linear regression of the recorded
arrival time ($t_a$) of the ion clouds at the detector versus the reciprocal drift voltage:

$$t_a = \frac{L_d^2}{K}\,\frac{1}{V_d} + t_0 \tag{3}$$

Note that the arrival time was determined from the centroid of the best-fit Gaussian
distribution, see Figure S2 in the Supplement. The $y$-intercept of the best-fit line
represents the transport time of the ion from the exit of the drift tube to the MS detector
($t_0$), which exhibits strong $m/z$ dependency that is attributable to a time-of-flight
separation in the ion optics, see Figure S3 in the Supplement.
It is practical to discuss an ion's mobility in terms of the reduced mobility constant
($K_0$), defined as:

$$K_0 = K\,\frac{273.15}{T}\,\frac{P}{1013.25} \tag{4}$$

where $P$ ($mbar$) is the pressure in the drift region and $T$ (K) is the buffer gas temperature.
In theory, the parameter $K_0$ is constant for a given ion in a given buffer gas and can be
used to characterize the intrinsic interactions of that particular ion-molecule pair. In
practice, however, $K_0$ values from different measurements might not be in good





agreement, primarily due to uncertainties in instrumental parameters such as
inhomogeneities in drift temperature and voltage (Fernández-Maestre et al., 2010). In
view of these uncertainties, the instrument standard (the reduced mobility of such a
standard is not affected by contaminants in the buffer gas) is needed to provide an
accurate constraint on the instrumental parameters, such as voltage, drift length, pressure,
and temperature:
$$K_0 \times t_d = \frac{L_d^2}{V_d} \frac{P}{1013.25} \frac{273.15}{T} = C_i \tag{5}$$

Tetraethyl ammonium chloride (TEA) is used here as the instrument standard
(Fernández-Maestre et al., 2010). Given the well-known $K_0$ and measured $t_d$ of the
protonated TEA ion ($m/z$ = 130), Equation (5) yields an instrument constant $C_i$ to
calibrate the IMS performance.
Unlike TEA, the reduced mobility of species that are more likely to cluster with
contaminants can be significantly affected by impurities of the buffer gas. This category
of species can be used as a 'mobility standard' to qualitatively indicate the potential
contamination in the buffer gas. 2,4-Lutidine, with a well-characterized $K_0$ value of 1.95
cm$^2$ V$^{-1}$ s$^{-1}$, is used as such a mobility standard. As shown Figure S4 in the Supplement,
the measured mobility of 2,4-Lutidine is 1.5% lower than its theoretical value, indicative
of the absence of contaminations in the buffer gas.
In the low field limit, the collision cross section of an ion ($\Omega$) with a buffer gas is
related to its reduced mobility ($K_0$) through the modified zero field (so called Mason-
Schamp) equation (McDaniel and Mason, 1973; Siems et al., 2012):
$$\Omega = \frac{3ze}{16N_0} \left(\frac{2\pi}{k_B \mu T_0}\right)^{1/2} \frac{1}{K_0} \left[1 + \left(\frac{\beta_{MT}}{\alpha_{MT}}\right)^2 \left(\frac{v_d}{v_T}\right)^2\right]^{-1/2} \tag{6}$$

where $z$ is the net number of integer charges on the ion, $e$ is the elementary charge, $N_0$ is
the number density of buffer gas at 273 $K$ and 1013 $mbar$, $k_B$ is the Boltzmann constant,
$\mu$ is the reduced mass for the molecule-ion pair, $T_0$ is the standard temperature, $v_d$ is the
drift velocity given by Equation (1), $v_T$ is the thermal velocity, and $\alpha_{MT}$ and $\beta_{MT}$ are
correction coefficients for collision frequency and momentum transfer, respectively,
given by:



$$\alpha_{MT} = \frac{2}{3}[1 + \hat{m}f_c + \hat{M}f_h] \qquad \beta_{MT} = [\frac{2}{\hat{m}(1+\hat{m})}]^{1/2} \qquad (7)$$

where $\hat{m}$ and $\hat{M}$ are mass fractions of the ion and buffer gas molecule, respectively, and $f_c$
and $f_h$ are the fractions of collisions in the cooling and heating classes, respectively. Note
that the reduced electric field used in this study is maximized at $\sim 2$ Td, at which the drift
velocity of any given ion is $\sim$ two orders magnitude lower than its thermal velocity, thus
the values for $f_c$ and $f_h$ are assigned to be 0.5 and 0.5, respectively. As all measurements
in this study were carried out with nitrogen as the buffer gas, the reported collision cross
sections will be referred to $\Omega_{N_2}$. Experimental $\Omega_{N_2}$ values for a selection of ionic species
are consistent with those reported in literatures (see Table S1 in the Supplement).

**3. Collision Cross Section Modeling**
Kinetic theory indicates that the quantity $\Omega$ is an orientationally averaged collision
integral ($\Omega_{avg}^{(l,l)}$), which depends on the nature of ion-neutral interaction potential
(McDaniel and Mason, 1973). Given the potential, the collision integral can be calculated
through successive integrations over collision trajectories, impact parameters and energy.
Here we adopt two computational methods, i.e., trajectory method and core model, to
simulate the average collision integral. The trajectory method is a rigorous calculation of
$\Omega_{avg}^{(l,l)}$ by propagating classical trajectories of neutral molecules in a realistic neutral/ion
potential consisting of a sum of pairwise Lennard-Jones interactions and ion induced
dipole interactions (Mesleh et al., 1996; Shvartsburg and Jarrold, 1996). The core model
treats the polyatomic ion as a rigid sphere where the center of charge is displaced from
the geometry center. The ion-neutral interaction is approximately represented by the cross
section of two rigid spheres during elastic collisions. The (12,4) potential, which includes
a long-range polarization term and a short-range repulsion term, is incorporated in the
core model (Mason et al., 1972).
The two models employed here represent opposite directions in the $\Omega_{avg}^{(l,l)}$ computation
methods. The trajectory method is a rigorous calculation of $\Omega_{avg}^{(l,l)}$ in a realistic
intermolecular potential yet the computation is time consuming. The core model, on the





other hand, substantially simplifies the calculation of $\Omega_{avg}^{(l,l)}$ as rigid sphere collisions at the
expense of simulation accuracy. We will show shortly that the core model is used for
locating individual chemical classes on the 2-D $\Omega_{N_2} - m/z$ space. Within the band
developed by each chemical class, molecular structure information can be deduced by
comparing the measured collision cross section with those calculated by the trajectory
method.
3.1 Trajectory Method
Molecular structures for L-leucine and D-isoleucine were initially constructed by
Avogadro v1.1.1 (Hanwell et al., 2012). For each molecule, both protonation and
deprotonation sites are created by placing a positive charge on the N-terminal amino
group and a negative charge on the C-terminal carboxyl group, respectively. The
geometry of each ion is further optimized using the Hartree-Fock method with the 6-
31G(d,p) basis set via GAMESS (Schmidt et al., 1993). Partial atomic charges were
estimated using Mulliken population analysis.
A freely available software, MOBCAL, developed by Jarrold and coworkers
(http://www.indiana.edu/~nano/software.html) was used for computing the collision
integrals. The potential term employed in the trajectory method takes the form:
$$\Phi(\theta,\phi,\gamma,b,r) = 4\epsilon\sum_i^n\left[\left(\frac{\sigma}{r_i}\right)^{12} - \left(\frac{\sigma}{r_i}\right)^6\right] - \frac{\alpha_p}{2}\left(\frac{ze}{n}\right)^2\left[\left(\sum_i^n\frac{x_i}{r_i^3}\right)^2 + \left(\sum_i^n\frac{y_i}{r_i^3}\right)^2 + \left(\sum_i^n\frac{z_i}{r_i^3}\right)^2\right] \qquad (8)$$
where $\theta$, $\phi$, and $\gamma$ are three angles that define the geometry of ion-neutral collision, $b$ is
the impact parameter, $\epsilon$ is the depth of the potential well, $\sigma$ is the finite distance at which
the interaction potential is zero, $\alpha_p$ is the polarizability of the neutral, which is $1.710 \times 10^{-24}$
$cm^3$ for $N_2$ (Olney et al., 1997), $n$ is the number of atoms in the ion, and $r_i$, $x_i$, $y_i$, and $z_i$
are coordinates that define the relative positions of individual atoms with respect to the
buffer gas. Values of the Lenard-Jones parameters, $\epsilon$ and $\sigma$, are taken from the universal
force field (Casewit et al., 1992). The ion-quadruple interaction and the orientation of $N_2$
molecule are not considered here (Kim et al., 2008; Campuzano et al., 2012).



### 3.2 Core Model

The core model, consisting of a (12-4) central potential displaced from the origin, is used to represent interactions of polyatomic ions with $N_2$ molecules (Mason et al., 1972). The central potential includes the common long-range $r^{-4}$ polarization energy, as well as the short-range $r^{-12}$ overlap repulsion energy:

$$\Phi(r) = \frac{\epsilon}{2} \left\{ \left( \frac{r_{\mathrm{m}} - a}{r - a} \right)^{12} - 3 \left( \frac{r_{\mathrm{m}} - a}{r - a} \right)^4 \right\} \tag{9}$$

where $r$ is the distance between the ion-neutral geometric centers, $a$ is the location of the ionic center of charge measured from the geometrical center of the ion, and $r_{\mathrm{m}}$ is the value of $r$ at the potential minimum. At temperature of 0 K, the *polarization potential* can be expressed as:

$$\Phi_{\mathrm{pol}}(r) = -\frac{e^2 \alpha_{\mathrm{p}}}{2r^4} \tag{10}$$

where $\alpha_{\mathrm{p}}$ is the polarizability of the neutral. Thus $\epsilon$ is given by:

$$\epsilon = \frac{e^2 \alpha_{\mathrm{p}}}{3(r_{\mathrm{m}} - a)^4} \tag{11}$$

The collision cross section can be expressed in dimensionless form by extracting its dependence on $r_{\mathrm{m}}$:

$$\Omega = \Omega^{(1,1)*} \pi r_{\mathrm{m}}^2 \tag{12}$$

Tabulations of the dimensionless collision integral ($\Omega^{(1,1)*}$) can be found in literatures (Mason et al., 1972) as a function of dimensionless temperature ($T^*$) and core diameter ($a^*$), given by:

$$T^* = \frac{kT}{\epsilon} = \frac{3kT(r_{\mathrm{m}} - a)^4}{e^2 \alpha_{\mathrm{p}}} \qquad a^* = \frac{a}{r_{\mathrm{m}}} \tag{13}$$

Polynomial interpolation of the tabulated $\Omega^{(1,1)*}$ yielded an analytical expression of the collision cross section, with $r_{\mathrm{m}}$ and $a$ as adjustable parameters. This expression was then fit to the ion mobility datasets measured in $N_2$ buffer gas using a nonlinear least-square



regression procedure (Matlab code is available upon request) (Johnson et al., 2004; Kim
et al., 2005; Kim et al., 2008). Best-fit parameters, $r_m$ and $a$, along with predicted vs.
measured collision cross section are given in Table S2 in the Supplementary Information.

**4. Collision Cross Section vs. Mass-to-Charge Ratio 2-D Space**
4.1 Distribution of *multi*-Functional Organic Species
Figure 1 (A) shows the distribution of organic species, classified as (*di/poly/sugar*)-
alcohol, *tertiary*-amine, *quaternary*-ammonium, (*mono/di*)-carbonyl, (*mono/di/tri*)-
carboxylic acid, (*di*)-ester, organic sulfate, and *multi*-functional compounds, on the
collision cross section vs. mass-to-charge ratio ($\Omega_{N_2} - m/z$) 2-D space. Note that analytes
that are detected in different ion modes (+/−) are plotted separately. One feature of the
distribution pattern is that species with higher density as pure liquids and carbon
oxidation state tend to occupy the lower region of the $\Omega_{N_2} - m/z$ space. This is not
surprising given that molecules of smaller collision cross sections tend to be much
denser, and potentially more functionalized, than those with extended and open
geometries. Furthermore, species of the same chemical class tend to occupy a narrow
region and follow a trend line on the $\Omega_{N_2} - m/z$ space. These observations form the basis
of potentially utilizing locations and trends on the 2-D space to identify chemical classes
to which an unknown compound belongs.
Small molecules ($m/z < 200$) with similar size and geometry are situated closely
together, as visualized by the 'overlaps' on the space. Improved visual separation of the
species within the overlapping region is obtained by transforming $\Omega_{N_2}$ to a quantity
$\Delta\Omega_{N_2}$, defined as the percentage difference between the measured collision cross section
for any given molecular ion and the calculated projection area for a rigid spherical ion-$N_2$
pair with the same molecular mass. A density of 1.2 g cm$^{-3}$, which represents the average
bulk density of ambient organic aerosol (Turpin and Lim, 2001), is used as the reference
value. Since this idealized ion-$N_2$ pair does not account for interaction potentials and
molecular conformation, it is only used as a reference state to improve visualization of
the $\Omega_{N_2} - m/z$ 2-D space, as shown in Figure 1 (B).



It is important to note that if these 'overlapping' molecules belong to different
chemical classes, they can be resolved based on the fragmentation pattern or
characteristic fragments upon collision induced dissociation (CID), as discussed in detail
in Section 4.3. Also note that isomeric and isobaric species can be identified by
comparing the measured collision cross sections with those obtained from trajectory
method simulations, see Section 4.4. As it is highly unlikely that two distinct molecules
will produce identical IMS, MS, as well as CID-based MS spectra, the 2-D framework
therefore virtually ensures reliable identification of compounds.
Reactions involving changes in functionalization and fragmentation can be represented
by an intrinsic directionality on the $\Omega_{N_2} - m/z$ space, as illustrated by the distribution
pattern of carboxylic acid series shown in Figure 2. Addition of one carbon always leads
to an increase in mass and collision cross section, with a generic slope of approximately 5
$\text{Å}^2/\text{Th}$. Although the addition of one oxygen in the form of a carbonyl group results in a
similar increase in the molecular mass, it leads to a shallower slope compared with that
from expanding the carbon chain. Addition of carboxylic or hydroxyl groups leads to a
substantial decrease in the collision cross section, due to the formation of a cyclic
conformation by the intramolecular hydrogen bonding $(O - H \cdots O^-)$.
4.2 $\Omega_{N_2} - m/z$ Trend Lines
The $\Omega_{N_2} - m/z$ trend line visualized on the 2-D space describes the intrinsic increase
in collision cross sections resulting from the increase in molecular mass by extending the
carbon backbone or adding functional groups. It has been used for conformation space
separation of different classes of biomolecules including lipids, peptides, carbohydrates,
and nucleotides (McLean et al., 2005). Here we demonstrate for the first time the
presence of trend lines for small molecules of atmospheric interest, and the trend line
pattern for each chemical class can be predicted by the core model simulations.
Figure 3 shows the measured $\Omega_{N_2}$ as a function of mass-to-charge ratio for (A)
*tertiary*-amine and *quaternary*-ammonium, (B) (*di/poly/sugar*)-alcohol, and (C)
(*mono/oxo/hydroxy*)-carboxylic acid. Also shown are the predicted $\Omega_{N_2}$ by the core
model, with adjustable parameters optimized by the measured $\Omega_{N_2}$ for the subcategory



spanning the largest *m/z* range in each chemical class. Specifically, *quaternary*-
ammonium, propylene glycol, and *alkanoic*-acid are used in constraining the core model
performance to predict the $\Omega_{N_2} - m/z$ trend lines for amines, alcohols, and carboxylic
acids. Species in each chemical class, regardless of the variety in the carbon skeleton
structure, occupy a narrow range and appear along a $\Omega_{N_2} - m/z$ trend line. Such a
relationship can be further demonstrated by the goodness of the core model predictions,
i.e., the difference between predicted and measured $\Omega_{N_2}$ for compounds that are not used
to optimize the core model performance. For amine series, predicted $\Omega_{N_2}$ values for
lutidine and pyridine are 8.2% and 0.8% higher, respectively, than the measurements. For
alcohol series, the best-fit $\Omega_{N_2} - m/z$ trend line constrained by propylene glycol can be
used to predict the distribution of sugars and polyols within 3.5% difference on the space.
For carboxylic acid series, hydroxyl-hexadecanoic acid falls closely on the predicted
$\Omega_{N_2} - m/z$ trend line, despite the presence of an alcohol group on the $C_{16}$ carbon chain.
Predicted $\Omega_{N_2}$ values for *oxo*-carboxylic acids are 4.4% – 6.1% lower than the
observations. Benzoic acid exhibits a relatively large measurement-prediction gap (6.7%)
potentially due to the presence of an aromatic ring.
Overall, the demonstrated $\Omega_{N_2} - m/z$ trend lines for carboxylic acids, amines, and
alcohols provide a useful tool for classification of structurally related compounds on the
space. It is worth noting that the core model optimization and simulation can be certainly
extended to other functionalities with the availability of chemical standards. Mapping out
the locations and distribution patterns for various functionalities on the 2-D space would
greatly facilitate structural identification of unknown compounds of atmospheric interest.
4.3 Molecular Structure Elucidation of *multi*-Functional Species
The demonstrated $\Omega_{N_2} - m/z$ relationship provides a useful tool to identify the
chemical class to which an unknown species belongs. To further identify its molecular
structure, knowledge on the electrospray ionization mechanism for the generation of
*quasi*-molecular ions, as well as fragmentation patterns of the molecular ion upon
collision induced dissociation (CID), is required.



For species investigated in this study, the positive mass spectra collected for amines
and amino acids show major ions at *m/z* values corresponding to the protonated cations
($[M+H]^+$). Sodiated clusters ($[M+Na]^+$) of esters were observed as the dominant peak in
the ESI(+) spectra. Aromatic aldehydes combine with a methyl group ($[M+CH_3]^+$) via the
gas-phase aldol reaction between protonated aldehydes and methanol in the positive
mode. Sugars and polyols can be readily ionized in both positive and negative mode with
the addition of a proton or sodium ion or deprotonation. Extensive formation of
oligomers is observed from the positive mass spectra of propylene glycol, with the
deprotonated propanol ($-OCH_2CH(CH_3)-$) as the primary building block. Monoanions
($[M-H]^-$) were exclusively observed in the negative mass spectra of (*mono/di/tri/multi*)-
carboxylic acids due to the facile ionization afforded by the carboxylic group. An
exhibition of molecular formulas of ionic species is given in Table 1.
The instrument used in this study enables the collision induced dissociation of the
abovementioned precursor ions after ion mobility separation but prior to the mass
spectrometer (IMS-CID-MS). As a consequence, product ions exhibit the identical
mobility (drift time) with that of the precursor ion. IMS-CID-MS spectra for individual
compounds are then generated by the extraction of "mobility-selected" MS spectra that
contain both precursor and fragments. The major advantage of this approach is that it is
possible to obtain fragmentation spectra for all precursor ions simultaneously. This is in
contrast to MS/MS techniques which require the isolation of a small mass window prior
to fragmentation which can be a problem for very complex samples or time-resolved
analysis. Figure 4 shows the measured drift time for the precursor and product ions
generated from species representative of amines, aldehydes, carboxylic acids, esters, and
nitro compounds. Collision induced dissociation patterns of these species are used to
elucidate the fragmentation mechanisms for corresponding functional groups. The
deprotonated carboxylic acid is known to undergo facile decarboxylation to produce a
carbanion. If additional carboxylic groups are present in the molecule, combined loss of
water and carbon dioxide is expected (Grossert et al., 2005). Alternatively, the presence
of an –OH group adjacent to the carboxylic group would usually result in a neutral loss of
formic acid (Greene et al., 2013), see the fragmentation pattern for 16-
hydroxyhexadecanoic acid as an illustration. Scission of the C–O bond in the ester





structure or the C–O bond between the secondary/tertiary carbon and the alcoholic
oxygen is observed for the ester series examined, consistent with previous studies (Zhang
et al., 2015). A primary fragmentation resulting in loss of CO was evident in the spectrum
of methylate derivative of protonated carbonyls ($RCHOCH_3^+$) (Neta et al., 2014). The
IMS-CID-MS spectrum of deprotonated 4-nitrophenol is shown as a representative of
organic nitro compounds. Two dominant peaks at *m/z* 108 and *m/z* 92 are observed,
resulting from the neutral loss of NO and $NO_2$, respectively.

Signal intensities of the fragments from the CID pathway of the precursor ion depend

on the collision voltage, as shown in Figure 5. At low collision voltages, the precursor
ions predominate with transmission optimized at approximately 5 V potential gradient.
As the collision voltage increases, the intensity of the precursor ion decreases and that of
each product ion increases, eventually reaching a maximum level, and then decreases due
to subsequent fragmentation. The dependence of the product ion abundance on the
collision voltage provides information on the relative strength of the covalent bond at
which the parent molecule fragments. Consequently, the energy required to induce a
certain fragmentation pathway could potentially also serve as an additional parameter for
structure elucidation. For example, the predominance of the product ion at m/z 149
suggests that cleavage of the O-O bond in the ester moiety is the dominant fragmentation
pathway upon CID of dioctyl phthalate ($C_{24}H_{38}O_4$).

4.4 Resolving Isomeric Mixtures

Here we demonstrate the separation and identification of isomers on the $\Omega_{N_2} - m/z$

space using the mixture of L-leucine and D-isoleucine as an illustration. Leucine can be
directly ionized by electrospray in both positive and negative modes due to the presence
of amino and carboxyl groups. Figure 6 (A and B) shows a single peak that corresponds
to the protonated ($[M+H]^+$, *m/z* = 132) and deprotonated ($[M–H]^-$, *m/z* = 130) forms of
the leucine mixture, respectively, in the positive and negative MS spectra. Upon further
separation based on their distinct mobility in the $N_2$ buffer gas, the leucine mixture is
clearly resolved in the positive mode, while a broad peak is observed in the negative ion
mobility spectrum, see Figure 6 (C and D). Note that a typical IMS resolving power



($t/dt_{50}$) of 100 leads to a baseline separation of leucine isomers that differ by 0.3 ms in the
measured drift time. Figure 6 (E-H) shows the IMS spectra for individual leucine
isomeric configurations, which provide precise constraints for the peak assignment in the
leucine mixture. Also given here are the measured vs. predicted collision cross sections
for each isomer, with predictions lower by 3.3 ~ 6.9% compared with the measurements.
However, despite the underprediction, the model using trajectory method correctly
predicts the relative collision cross sections of the isomers and therefore also the order in
which they appear in the IMS spectrum. The underprediction of $\Omega_{N_2}$ may result from the
simplification that linear $N_2$ molecules are considered as elastic and specular spheres in
the current model configuration (Larriba-Andaluz and Hogan Jr, 2014). Further
development of the model to more appropriately predict $\Omega_{N_2}$ values is needed.

**5. Conclusions**

We propose a new metric, collision cross section ($\Omega$), for characterizing organic
species of atmospheric interest. Collision cross section represents an effective interaction
area between a charged molecule and neutral buffer gas as it travels through under the
action of a weak electric field, and thus relates to the chemical structure and 3-D
conformation of this molecule. The collision cross section of individual molecular ions is
calculated from the ion mobility measurements using an Ion Mobility Spectrometer. In
this study, we provide the derived $\Omega_{N_2}$ values for a series of organic species including
amines, alcohols, carbonyls, carboxylic acids, esters, organic sulfates, and *multi-*
*functional* compounds.
The collision cross section, when coupled with mass-to-charge ratio, provides a 2-D
framework for characterizing the molecular signature of atmospheric organic aerosol.
The $\Omega_{N_2} - m/z$ space is employed to guide our fundamental understanding of processes
of organic aerosol formation and evolution in the atmosphere. We show that different
chemical classes tend to develop unique narrow bands with trend lines on the $\Omega_{N_2} - m/z$
space. Trajectories associated with atmospheric transformation mechanisms either cross
or follow these trend lines through the space. The demonstrated $\Omega_{N_2} - m/z$ trend lines





provide a useful tool for resolving various functionalities in the complex OA mixture.
These intrinsic trend lines can be predicted by the core model, which provides a guide for
locating unknown functionalities on the $\Omega_{N_2} - m/z$ space.
Within each band that that belongs to a particular chemical class on the space, species
can be further separated based on their distinct structures and geometries. We
demonstrate the utility of collision induced dissociation technique, upon which the
resulted product ions share the identical drift time as the precursor ion, to facilitate the
elucidation of molecular structures of OA constituents. We employ the $\Omega_{N_2} - m/z$
framework for separation of isomeric mixtures as well by comparing the measured
collision cross sections with those predicted using the trajectory method. Further
advances in algorithms to correctly predict collision cross sections *ab initio* from
molecular coordinates are therefore also expected to significantly improve identification
of unknowns.

**Acknowledgement**
This study was supported by the U.S. National Science Foundation (NSF)
Atmospheric and Geospace Sciences (AGS) grants 1537446. J.E.K. was supported by
fellowships from CIRES and EPA STAR (FP-91770901-0). J.L.J. was supported by DOE
(BER/ASR) DE-SC0011105 and EPA STAR 83587701-0. This manuscript has not been
reviewed by EPA and thus no endorsement should be inferred.

**Appendix:**
$a$ (Å): the location of the ionic center of charge from the geometrical center of the ion.
$a^*$: the dimensionless core diameter.
$\alpha_{MT}$: the correction coefficient for collision frequency.
$\alpha_p$ (cm$^3$): the polarizability of the neutral.
$\beta_{MT}$: the correction coefficient for momentum transfer.
$C_i$: the instrument constant that is used to calibrate the IMS performance.
$\epsilon$ (eV): the depth of the potential well.





$E_d$ (V/cm): the electric field intensity in the drift tube.
$\Phi$ (eV): the ion-neutral interaction potential.
$f_c$: the fraction of collisions in the cooling classes.
$f_h$: the fraction of collisions in the heating classes.
$k_B$ (m$^2$ kg s$^{-2}$ K$^{-1}$): Boltzmann constant.
$K$ (cm$^2$ V$^{-1}$ s$^{-1}$): the characteristic mobility constant of a given ion.
$K_0$ (cm$^2$ V$^{-1}$ s$^{-1}$): the reduced mobility constant of a given ion.
$L_d$ (V/cm): the length of the drift tube.
$\hat{m}$: the mass fraction of the ion in the ion-molecule pair.
$\hat{M}$: the mass fraction of the buffer gas molecule ($N_2$) in the ion-molecule pair.
$m/z$ (Th): the mass-to-charge ratio of any given ion.
$N_0$ (# cm$^{-3}$): the number density of buffer gas at 273 $K$ and 1013 $mbar$.
$\Omega$ (Å$^2$): the collision cross section.
$\Omega_{N_2}$ (Å$^2$): the collision cross section using $N_2$ as the buffer gas.
$\Omega_{avg}^{(l,l)}$: the orientationally averaged collision integral.
$\Omega^{(l,l)*}$: the dimensionless collision integral.
$P$ ($mbar$): the pressure in the drift region.
$r$ (Å): the distance between the ion-neutral geometric centers.
$r_m$ (Å): the value of $r$ at the potential minimum.
$\sigma$ (Å): the finite distance at which the interaction potential is zero.
$T$ (K): the buffer gas temperature.
$T_0$ (K): the standard temperature.
$T^*$: the dimensionless temperature.
$t_a$ (s): the recorded arrival time of the ion clouds at the detector.
$t_d$ (s): the time of ion clouds spent in the drift tube.
$t_0$ (s): the transport time of ion clouds from the exit of the drift tube to the MS detector.
$v_d$ (s): the average velocity of an ion in the drift tube.
$v_T$ (m s$^{-1}$): the thermal velocity.
$V_d$ (V): the voltage applied to the drift tube.
$z$: the net number of integer charges on the ion.



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



Table 1. Overview of organic standards investigated in this study.

| Class | Chemical | Molecular Formula | Ion | | $\Omega_{N_2}$ (Å$^2$) | Molecular Structure |
|---|---|---|---|---|---|---|
| | | | Formula | m/z | | |
| Amine | Tetraethyl ammonium chloride | C$_8$H$_{20}$NCl | [M-Cl]$^+$ | 130.16 | 122.1 | |
| | Tetrapropyl ammonium chloride | C$_{12}$H$_{28}$NCl | [M-Cl]$^+$ | 186.10 | 143.8 | |
| | Tetrabutyl ammonium iodide | C$_{16}$H$_{36}$NI | [M-I]$^+$ | 242.17 | 165.8 | |
| | Tetrapentyl ammonium chloride | C$_{20}$H$_{44}$NCl | [M-Cl]$^+$ | 298.35 | 190.0 | |
| | Tetraheptyl ammonium chloride | C$_{28}$H$_{60}$NCl | [M-Cl]$^+$ | 410.47 | 236.5 | |
| | 2,4-Lutidine | C$_7$H$_9$N | [M+H]$^+$ | 108.08 | 123.4 | |
| | 2,6-Di-tert-butylpyridine | C$_{13}$H$_{21}$N | [M+H]$^+$ | 192.17 | 145.0 | |
| Amino acid | L-Leucine | C$_6$H$_{13}$NO$_2$ | [M+H]$^+$ [M–H]$^-$ | 132.10 130.09 | 137.8 144.4 | |
| | D-Isoleucine | C$_6$H$_{13}$NO$_2$ | [M+H]$^+$ [M–H]$^-$ | 132.10 130.09 | 135.2 140.3 | |
| mono Carboxylic Acid | Benzoic acid | C$_7$H$_6$O$_2$ | [M–H]$^-$ | 121.03 | 128.6 | |
| | Octanoic acid | C$_8$H$_{16}$O$_2$ | [M–H]$^-$ | 143.11 | 144.7 | |
| | 2-Butyloctanoic acid | C$_{12}$H$_{24}$O$_2$ | [M–H]$^-$ | 199.17 | 162.0 | |
| | Tridecanoic acid | C$_{13}$H$_{26}$O$_2$ | [M–H]$^-$ | 213.19 | 166.2 | CH$_3$(CH$_2$)$_{10}$CH$_2$ |
| | Pentadecanoic acid | C$_{15}$H$_{30}$O$_2$ | [M–H]$^-$ | 241.22 | 173.7 | CH$_3$(CH$_2$)$_{12}$CH$_2$ |
| | Palmitic acid | C$_{16}$H$_{32}$O$_2$ | [M–H]$^-$ | 255.23 | 177.9 | CH$_3$(CH$_2$)$_{13}$CH$_2$ |
| | Stearic acid | C$_{18}$H$_{36}$O$_2$ | [M–H]$^-$ | 283.26 | 185.4 | CH$_3$(CH$_2$)$_{15}$CH$_2$ |
| | Oleic acid | C$_{18}$H$_{34}$O$_2$ | [M–H]$^-$ | 281.25 | 186.9 | CH$_3$(CH$_2$)$_6$CH$_2$ |



| | | | | | | |
|---|---|---|---|---|---|---|
| | Succinic acid | $C_4H_6O_4$ | $[M–H]^-$ | 117.02 | 124.6 | |
| | Glutaric acid | $C_5H_8O_4$ | $[M–H]^-$ | 131.03 | 128.4 | |
| | Adipic acid | $C_6H_{10}O_4$ | $[M–H]^-$ | 145.05 | 131.5 | |
| | Pimelic acid | $C_7H_{12}O_4$ | $[M–H]^-$ | 159.06 | 134.0 | |
| | Azelaic acid | $C_9H_{16}O_4$ | $[M–H]^-$ | 187.09 | 143.5 | |
| *di/multi* Carboxylic Acid | Sebacic acid | $C_{10}H_{18}O_4$ | $[M–H]^-$ | 201.11 | 148.9 | |
| | 1,2,3-Propane tricarboxylic acid | $C_6H_8O_6$ | $[M–H]^-$ | 175.02 | 122.2 | |
| | Cyclohexane tricarboxylic acid | $C_9H_{12}O_6$ | $[M–H]^-$ | 215.06 | 135.0 | |
| | Mellitic acid | $C_{12}H_6O_{12}$ | $[M–H_2O–H]^-$ | 322.96 | 154.6 | |
| | Dibutyl oxalate | $C_{10}H_{18}O_4$ | $[M+Na]^+$ | 225.11 | 170.0 | |
| Ester | Dibutyl phtahlate | $C_{16}H_{22}O_4$ | $[M+Na]^+$ $[2M+Na]^+$ | 301.14 579.29 | 192.4 255.5 | |
| | Dioctyl phthalate | $C_{24}H_{38}O_4$ | $[M+H]^+$ | 391.28 | 203.6 | |
| Alcohol | Propylene glycol | $C_3H_8O_2$ | $[2M-2H_2O+Na]^+$ | 215.12 | 144.8 | |
| | | | $[4M-3H_2O+Na]^+$ | 273.17 | 156.4 | |
| | | | $[5M-4H_2O+H]^+$ | 309.23 | 165.7 | |
| | | | $[5M-4H_2O+Na]^+$ | 331.21 | 169.6 | |
| | | | $[6M-5H_2O+H]^+$ | 367.27 | 179.1 | |
| | | | $[6M-5H_2O+Na]^+$ | 389.24 | 181.6 | |
| | | | $[7M-6H_2O+H]^+$ | 425.31 | 190.8 | |
| | | | $[7M-6H_2O+Na]^+$ | 447.28 | 193.9 | |
| | | | $[8M-7H_2O+H]^+$ | 483.35 | 204.7 | |
| | | | $[8M-7H_2O+Na]^+$ | 505.32 | 206.2 | |
| | | | $[9M-8H_2O+H]^+$ | 541.39 | 218.5 | |
| | | | $[9M-8H_2O+Na]^+$ | 563.36 | 219.3 | |
| | | | $[10M-9H_2O+H]^+$ | 599.42 | 231.3 | |
| | | | $[10M-9H_2O+Na]^+$ | 621.40 | 231.8 | |



| | Name | Formula | Ion | m/z | | Structure |
|---|---|---|---|---|---|---|
| | DL-Threitol | $C_4H_{10}O_4$ | $[M+Na]^+$ | 145.05 | 133.0 | |
| | Xylitol | $C_5H_{12}O_5$ | $[M–H]^-$ | 151.06 | 131.2 | |
| | Sucrose | $C_{12}H_{22}O_{11}$ | $[M–H]^-$ $[M+Na]^+$ | 341.11 365.11 | 167.6 175.1 | |
| Carbonyl | Hexane-3,4-dione | $C_6H_{10}O_2$ | $[M+H]^+$ $[M+CH_3]^+$ | 115.08 129.09 | 115.7 121.3 | |
| | Acetophone | $C_8H_8O$ | $[M+CH_3]^+$ | 135.08 | 120.4 | |
| | Cinnamaldehyde | $C_9H_8O$ | $[M+CH_3]^+$ | 147.08 | 123.9 | |
| | Levulinic acid | $C_5H_8O_3$ | $[M–H]^-$ | 115.04 | 130.0 | |
| | 4-Acetylbutyric acid | $C_6H_{10}O_3$ | $[M–H]^-$ | 129.06 | 134.5 | |
| | Homovanillic acid | $C_9H_{10}O_4$ | $[M–H]^-$ | 181.05 | 147.7 | |
| *multi* Functional Compound | 16-Hydroxy hexadecanoic acid | $C_{16}H_{32}O_3$ | $[M–H]^-$ | 271.22 | 183.7 | |
| | Oxaloacetic acid | $C_4H_4O_5$ | $[M–H]^-$ | 131.06 | 118.3 | |
| | Ketoglutaric acid | $C_5H_6O_5$ | $[M–H]^-$ | 145.01 | 120.9 | |
| | Oxoazelaic acid | $C_9H_{14}O_5$ | $[M–H]^-$ | 201.08 | 133.3 | |
| | Malic acid | $C_4H_6O_5$ | $[M–H]^-$ | 133.01 | 111.4 | |
| | Tartaric acid | $C_4H_6O_6$ | $[M–H]^-$ | 149.01 | 116.0 | |
| | Citric acid | $C_6H_8O_7$ | $[M–H]^-$ | 191.02 | 123.0 | |
| Organic Sulfate | Sodium Dodecyl sulfate | $C_{12}H_{25}SO_4Na$ | $[M–Na]^-$ | 265.15 | 163.6 | |
| | Sodium Taurocholate | $C_{26}H_{44}SO_7NNa$ | $[M–Na]^-$ | 514.28 | 206.4 | |



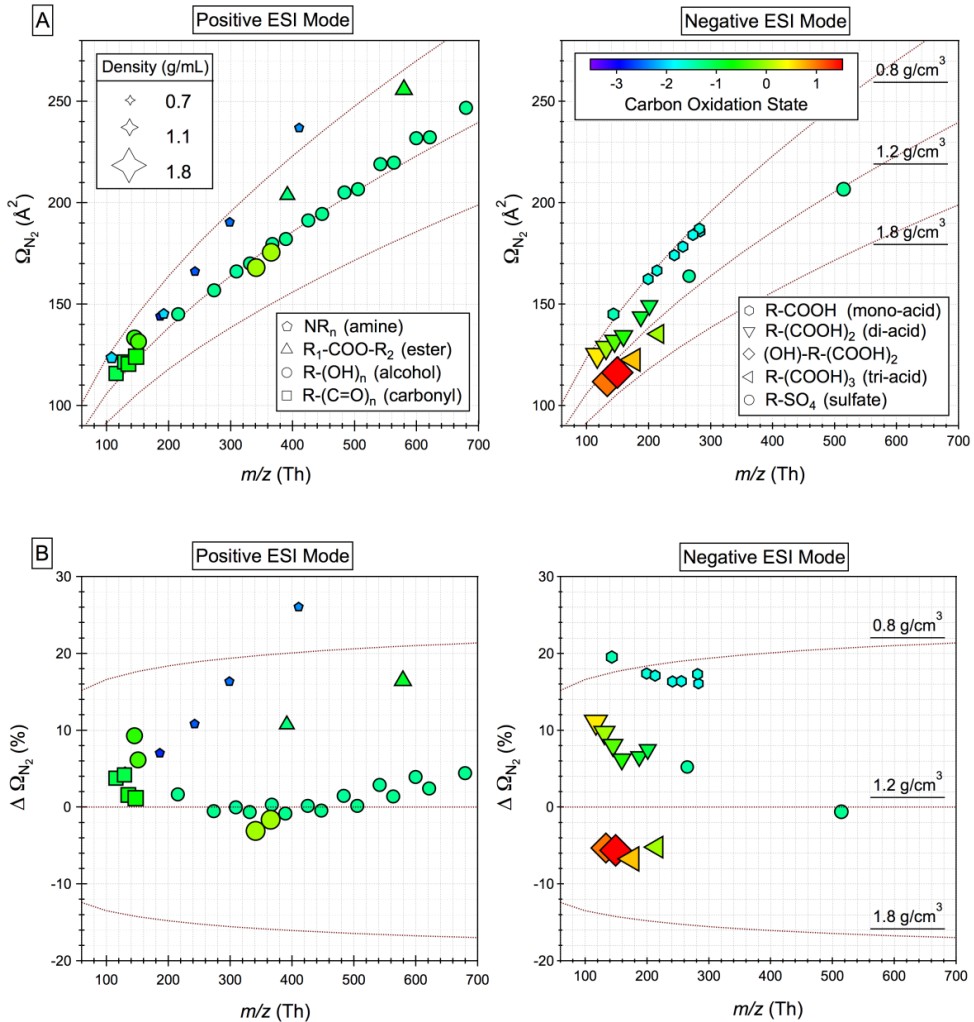

Figure 1. Distribution of organic species including alcohol (R-(OH)$_n$, $n$ = 2-8), amine (NR$_3$), *quaternary*-ammonium (NR$_4$), carbonyl (R-(C=O)$_n$, $n$ = 1-2), carboxylic acid (R-(COOH)$_n$, $n$ = 1-3), ester (R$_1$-COO-R$_2$), organic sulfate (R-SO$_4$), and *multi*-functional compounds ((OH)-R-(COOH)$_2$) on the (A) $\Omega_{N_2} - m/z$ space and (B) $\Delta\Omega_{N_2} - m/z$ space.





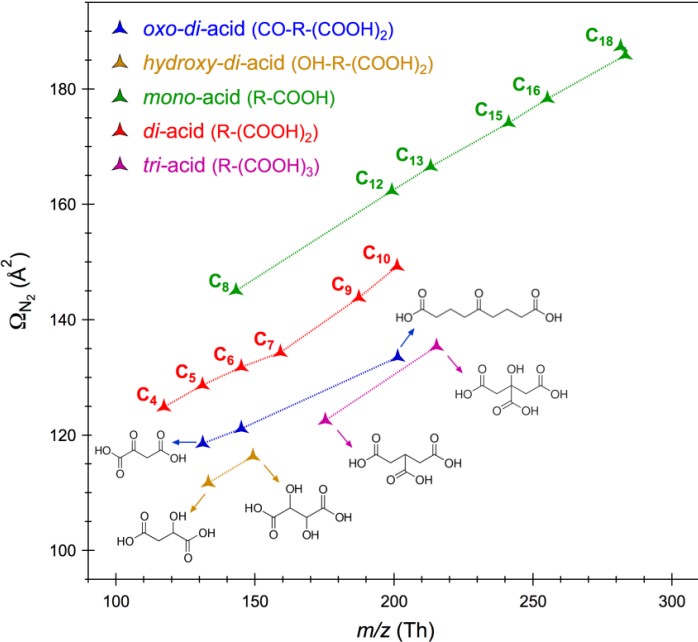

Figure 2. Trajectories associated with reactions involving functionalization (changes in the type and number of functional groups) and fragmentation (changes in the carbon chain length) through the 2-D $\Omega_{N_2} - m/z$ space using carboxylic acid series as an illustration.



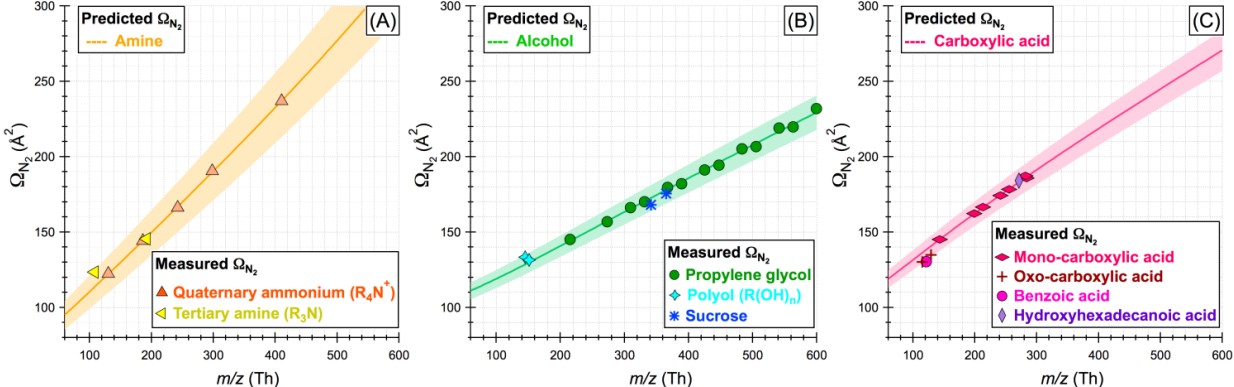

Figure 3. Measured collision cross sections ($\Omega_{N_2}$) for (A) *tertiary*-amine and *quaternary*-ammonium, (B) (*di/poly/sugar*)-alcohol, and (C) (*mono/oxo/hydroxy*)-carboxylic acid as a function of the mass-to-charge ratio. Also shown are the predicted $\Omega_{N_2} - m/z$ trend lines for amine, alcohol, and carboxylic acid by the core model. Here, *quaternary*-ammonium, propylene glycol, and $C_8$-$C_{18}$ *alkanoic*-acid are used to optimize the adjustable parameters in the core model (The markers are in the same color as the trend lines). The colored shade in each figure represents the maximum deviations (8.21%, 3.54%, and 6.69% for amine, alcohol, and carboxylic acid, respectively) of the predicted $\Omega_{N_2}$ from the measured $\Omega_{N_2}$ for species that are not used to constrain the core model. A single plot showing the separation of these three chemical classes is given in Figure S5 in the supplement.





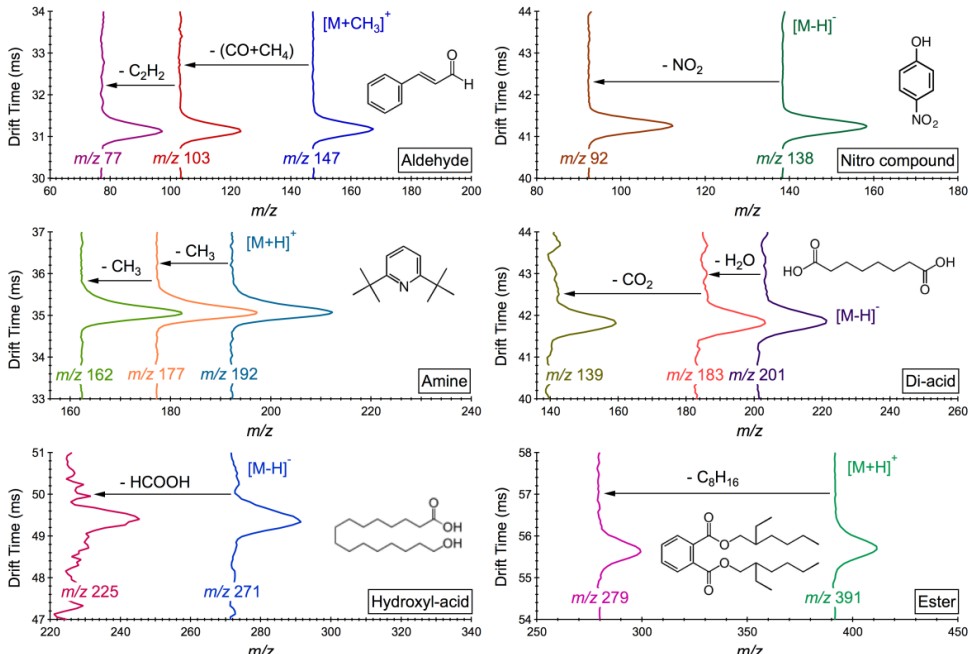

Figure 4. Collision induced dissociation patterns for molecular ions generated from cinnamaldehyde, dioctyl phthalate, 2,6-di-tert-butylpyridine, 4-nitrophenol, 16-hydroxyhexadecanoic acid, and sebacic acid on the mobility – mass framework with mass-to-charge ratio on the *x*-axis and drift time on the *y*-axis. The corresponding mobility selected MS spectra for each species is given in Figure S6 in the supplement.




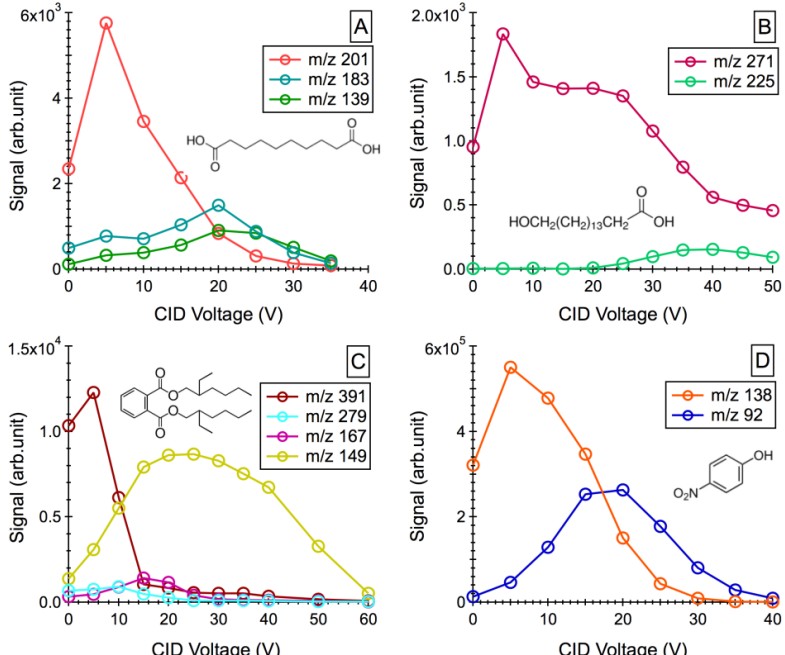

Figure 5. Product ion peak intensities as a function of collision voltage in the 'mobility-selected' MS spectra of (A) deprotonated sebacic acid, (B) deprotonated 16-hydroxyhexadecanoic acid, (C) sodiated dioctyl phthalate, and (D) deprotonated 4-nitrophenol.

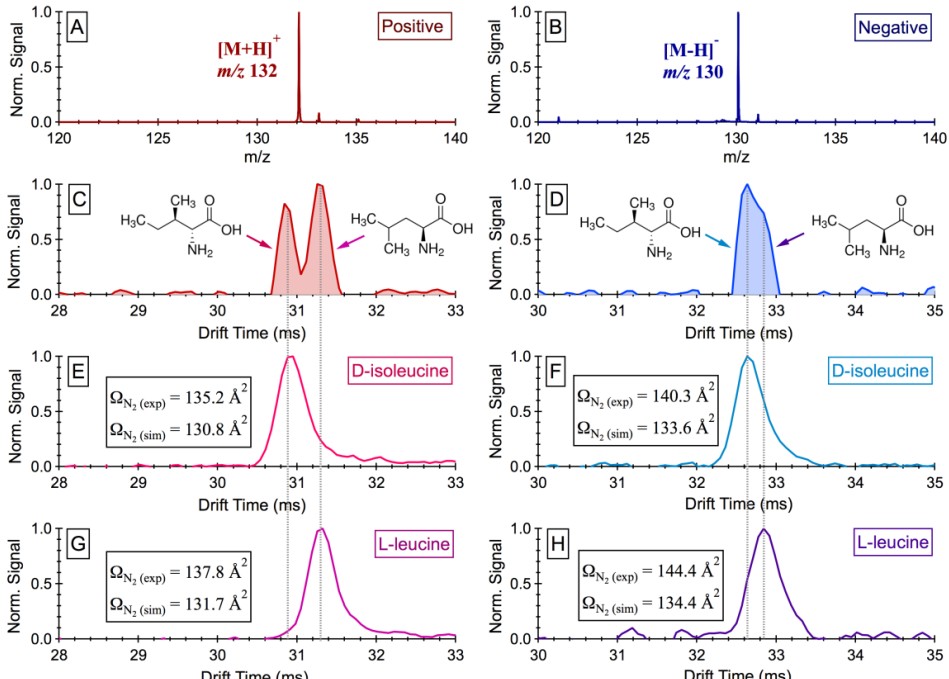

Figure 6. (A/B) ESI mass spectra collected for an equi-molar mixture (20 μM each) of L-leucine and D-isoleucine in positive and negative mode. (C/D) Measured drift time distributions for the leucine mixture in positive and negative mode. (E/F) Measured vs. predicted $\Omega_{N_2}$ for D-isoleucine, together with its drift time distributions in positive and negative mode. (G/H) Measured vs. predicted $\Omega_{N_2}$ for L-leucine, together with its drift time distributions in positive and negative mode. Note that all measurements were performed at ~ 303 K and ~ 1019 *mbar* with an electric field strength of 414 and 403 V cm$^{-1}$ in the positive and negative mode, respectively.