# Peer review of "A Novel Framework for Molecular Characterization of Atmospherically Relevant Organic Compounds Based on Collision Cross Section and Mass-to-Charge Ratio"

_Atmospheric Chemistry and Physics, 2016_

## Referee Comment (RC1) · Anonymous Referee #2 · 31 Jul 2016

In this work, Zhang et al. presented a new framework to describe complex organic mixture within aerosols. The authors first introduced the 2-dimensional cross section-m/z framework and methods for calculating ion-neutral collisional cross sections. They demonstrated the applicability using a range of standard compounds, and showed unique behaviors in the 2-D space. The authors also showed how molecular identification can be performed using collision-induced dissociation. This framework is novel and unique, and addresses an important knowledge gap in accounting for molecular structures/functional groups in the organic aerosols. The work is thorough and the manuscript is very well presented. I have some very minor concerns about how to apply this framework broadly, which are more about framing the work in a broader con-

text. This manuscript should be published in Atmospheric Chemistry and Physics after addressing these minor comments.

Major comments:

1. I am not quite sure how this framework would work with real atmospheric mixtures, which likely contain many multifunctional organic compounds. A very unique feature of this technique is the cross section decreases with increasing oxygenation. However, there are different types of oxygenated functional groups. For example, from Figure 2 it appears that ketone group lowers cross section the same amount as a 2nd carboxylic group does. One might not be able to identifying uniquely the structure of the molecules based solely on the location in this plane. Rather it is possible to identify the general trend (e.g. shifts in location as a function of time/oxidation) during oxidation or atmospheric processes. This is not a critique of the framework itself, but I would like to see a discussion of the limitations and/or applicability to better assess its useful for different purposes.

2. Along the same lines of limitations/applicability, lines 349-350 seem to suggest that aromatic compounds may exhibit significant deviations. In atmospheric mixtures, there will be a larger mix of aliphatic/aromatic compounds. Would that imply this framework will work well for laboratory experiments to constrain oxidation, but not for atmospheric mixtures?

3. In this work, the authors used electrospray ionization, and there are a number of problems with ESI. First the ionization chemistry is very complex. In fact, the authors dedicated a whole paragraph (lines 363 – 374) to explain the chemistry, and the complexity can also be seen in Table 1. While ESI is a universal technique and is able to ionize almost all molecules one would encounter in SOA, it will be difficult (or, at the very least, tedious) to work backwards and deduce the original molecules from the large set of ion formulas observed. The second problem is that ESI is not a quantitative technique, especially with direct infusion shown here without prior separation. ESI

suffers from matrix effects, and it is difficult to use surrogate standards for quantification. Perhaps the authors should point out to readers that more quantitative ionization techniques should be used to fully exploit the usefulness of the 2D framework.

4. Resolving isomeric structures: leucine and isoleucine are biological molecules and there are many other techniques that are capable for resolving those compounds. I find the use of leucine and isoleucine to demonstrate the capability of isomer separation to be not too effective. Perhaps the authors can consider showing the capability of the IMS to separate molecules of atmospheric interest?

Minor:

Figure 2: it seems to me that citric acid and the tricarboxylic acid are not quite the same (differs by an –OH group, the other homologue series differs by one or more –CH2- groups) so the trend line should not be drawn the same way as in the other series.

Figure 4: I don't understand the fragmentation pattern in the amine (di-tert-butyl pyridine). Why is there a loss of –CH3 group? (It will form an unstable ion with an unpaired electron.)

---

## Referee Comment (RC2) · Anonymous Referee #1 · 2 Aug 2016

In this manuscript, the authors present a new framework with which to identify and characterize atmospherically relevant organic compounds based on a combination of ion mobility and molecular mass. Though a wide variety of such two-dimensional frameworks have been employed to parameterize and simplify descriptions of atmospheric mixtures, the technique proposed by the authors is unique and valuable in its ability to characterize compounds based on structural features and to separate isomeric species. The authors have done an excellent and detailed job of exploring intrinsic spatial relationships in this parameter space. Continued application of the tools and techniques described in this manuscript will likely provide more molecular and chemical information than has previously been available.

General comments:

While this manuscript builds a strong foundation for the application of these techniques to atmospheric samples, no attempt is made to apply these techniques to complex mixtures of unknowns. The title, abstract, and some portions of the introduction should be re-framed to highlight what is actually in this manuscript and focus less on what the authors hope to do with these tools in the future. It is implied or, in the case of the title stated explicitly, that this paper is about the "Characterization of Atmospheric Organic Aerosol." Given the home institutions of the authors, I have no doubt this is the goal and am excited to see it applied to ambient mixtures. However, without more detailed attempts to apply this technique to atmospheric mixtures or at least detailed discussion, the language and title of this manuscript should be changed to focus more on "atmospherically relevant organic compounds," or "highly oxidized small organic compounds," or "characterizing functionality of organic compounds."

The detailed description of the framework also needs some added clarity – see detailed comments below.

Minor comments:

Line 40: "scatter plots" is not a verb

Line 64: Re-word, perhaps use "and subsequent interactions" in place of "as well as"

Line 169: This phrase is awkwardly broken up and should be re-worded: "the instrument standard (the reduced mobility of such a standard is not affected by contaminants in the buffer gas) is needed"

Lines 184-200 describe the apparent crux of this framework, but some points could be made clearer. In particular, explicitly relating the measured parameters to the calculated parameters would be very helpful. For instance, in Eq. 6, it would be useful to re-frame in terms of $t\_d$ since that is what is actually being measured, instead of $K\_0$ and $v\_d$. What is the functional form of this relationship, considering all of the terms in

the equation? Is it collision cross section generally linear with drift time? Relatedly: • Line 187: Is z=1 assumed for all ions? In contrast to other ionization techniques, ESI can under some conditions yield a distribution of charges – is this an issue and to what extent would it change the results? • Line 190: Is thermal velocity calculated as a function of molecular mass? • Line 193: How are the mass fractions calculated? How might this work for a mixture of unknowns with poorly defined sensitivities?

Line 214: Define or clarify "(12,4) potential"

Sections 3.1 and 3.2 could perhaps be switched, as they are discussed in Section 3 and in Section 4 in the opposite order.

Line 301: It is not clear to me that "functionalization and fragmentation can be represented by an intrinsic directionality" as claimed by the authors. As the authors note, the connected markers shown in Figure 2 represent addition of non-functionalized carbon atoms, but this is not the form that atmospheric functionalization takes. Instead, addition of carbon moves up and to the right, but addition of functional groups appears to move down. What would a vector of functionalization or fragmentation look like in this space? This question is particularly important if the authors intend to keep their focus on using this approach to characterize complex mixtures.

Line 345: While the trend lines described in the Section 4.2 provide an important characterization of this framework, and a useful test of the core model, it's not completely clear they would be particularly help in identifying unknown species, as implied in this sentence. As demonstrated by Figure S5, there is substantial overlap between the regions of all of the trend lines – if one were handed an unknown, its location in this space alone would not provide much information on its family. These lines are perhaps useful for identifying family of species, and demonstrate the utility of the core model for helping to understand its location in the space, but as written this sentence is a bit of an overstatement without some explanation or support. Section 4.3, on the other hand, does indeed seem very promising for identifying unknowns. . .

Line 386: A dominant peak at 108 is mentioned but not shown in Figure 4.

Line 398: The there is no O-O bond in dioctyl phathalate. Do the authors mean the carbonyl-oxygen bond? Interestingly (and relatedly), this 149 peak is the dominant peak in EI spectra.

---

## Author Comment (AC1) · 21 Sep 2016

**Response to Reviewer #2**

In this work, Zhang et al. presented a new framework to describe complex organic mixture within aerosols. The authors first introduced the 2-dimensional cross section-m/z framework and methods for calculating ion-neutral collisional cross sections. They demonstrated the applicability using a range of standard compounds, and showed unique behaviors in the 2-D space. The authors also showed how molecular identification can be performed using collision-induced dissociation. This framework is novel and unique, and addresses an important knowledge gap in accounting for molecular structures/functional groups in the organic aerosols. The work is thorough and the manuscript is very well presented. I have some very minor concerns about how to apply this framework broadly, which are more about framing the work in a broader context. This manuscript should be published in Atmospheric Chemistry and Physics after addressing these minor comments.

*We thank reviewer #2 for the constructive and insightful comments. Our point-by-point responses can be found below, with reviewer comments in **black**, our responses in **blue**, alongside the relevant revisions to the manuscript in **red**.*

Major comments:

1. I am not quite sure how this framework would work with real atmospheric mixtures, which likely contain many multifunctional organic compounds. A very unique feature of this technique is the cross section decreases with increasing oxygenation. However, there are different types of oxygenated functional groups. For example, from Figure 2 it appears that ketone group lowers cross section the same amount as a 2nd carboxylic group does. One might not be able to identifying uniquely the structure of the molecules based solely on the location in this plane. Rather it is possible to identify the general trend (e.g. shifts in location as a function of time/oxidation) during oxidation or atmospheric processes. This is not a critique of the framework itself, but I would like to see a discussion of the limitations and/or applicability to better assess its useful for different purposes.

**[Responses]** The reviewer has raised a very important issue on utilizing the $\Omega - m/z$ space to resolve complex aerosol mixture, and we agree that it is rather difficult to identify uniquely the structure of the molecules based solely on the location of unknowns on the $\Omega - m/z$ space. Therefore, we highlight the use of Collision Induced Dissociation (CID) in *Section 4.4 'Molecular Structure Elucidation of multi-Functional Species'* as an additional dimension of

separation. We also emphasize that the CID method becomes crucial if the locations of two unknown species cannot be well resolved on the space under the current IMS resolution ($t/dt \sim$ 100). The key message we would like to deliver in this manuscript is that a combined knowledge on the mass to charge ratio of the unknowns, the location of the unknowns on the 2-D space, as well as the fragmentation patterns of unknowns, could provide information on the chemical classes to which the unknowns belong as well as the functional groups the unknowns contain. We have refined the major conclusions in the revised manuscript, also given below:

[Changes] Line 318-328: The demonstrated $\Omega_{N_2} - m/z$ trend lines provide a useful tool for categorization of structurally related compounds. Mapping out the locations and distribution patterns for various functionalities on the 2-D space would therefore facilitate classification of chemical classes for unknown compounds. It is likely that trend lines extracted from a complex organic mixture overlap and, as a result, the distribution pattern of unknowns on the space alone would not provide sufficient information on their molecular identities. In this case, the fragmentation pattern of unknowns upon collision induced dissociation (CID) needs to be explored for the functionality identification, as discussed in detail in Section 4.4. As it is highly unlikely that two distinct molecules will produce identical IMS, MS, as well as CID-based MS spectra, the 2-D framework therefore virtually ensures reliable identification of species of atmospheric interest.

[Responses] Another important issue raised by the reviewer is the complexity of atmospheric aerosol mixtures. In this manuscript, we have characterized alcohols, amines, aldehydes, carbonyls, carboxylic acids, esters, and organic sulfates. We note that we have devoted our recent efforts to the characterization of more chemical classes that are atmospheric interest, including two families that are representative of rural and urban atmospheres, respectively, i.e., organic peroxides and nitrates. The goal is to fill more information on the space to facilitate the investigation of complex organic aerosol mixture.

[Responses] Work in progress has been evaluating the application of the IMS-MS technique to laboratory generated SOA mixtures. The application of IMS-MS to a rather simple SOA system that is generated from the reactive uptake of IEPOX onto acidified ammonium sulfate seed particles, for example, has been demonstrated by our recent study (Krechmer et al. AMT, 2016). Figure 5c in Krechmer et al. (2016) shows the distribution of IEPOX derived organosulfate, along with its dimers and trimers, on the 2-D space. A trend line for the major IEPOX monomers, dimers, and trimmers is clearly visible. The characteristic fragment upon CID, i.e., sulfate (m/z 97), adds further confirmation on the chemical identities of the IEPOX derived products in the particle phase.

2. Along the same lines of limitations/applicability, lines 349-350 seem to suggest that aromatic compounds may exhibit significant deviations. In atmospheric mixtures, there will be a larger mix of aliphatic/aromatic compounds. Would that imply this framework will work well for laboratory experiments to constrain oxidation, but not for atmospheric mixtures?

**[Responses]** Figure 3(C) in the manuscript shows that benzoic acid deviates by ~ 8% from the predicted trend lines for alkanoic acids. If this deviation is attributed to the presence of the planar aromatic ring, one would expect that all aromatics exhibit deviations in the measured $\Omega_{N_2}$ from the aliphatic ones, even if they contain the same type and number of functional groups. For example, within the narrow band of *mono*-carboxylic acids, there would be two sub-lines that characterize the aromatic series and aliphatic series, respectively. We think the reviewer raises a good point and we need further studies to refine these trend lines for aromatics. Collected filter samples for SOA generated from the photooxidation of aromatic precursors will be a good start to provide relevant information.

3. In this work, the authors used electrospray ionization, and there are a number of problems with ESI. First the ionization chemistry is very complex. In fact, the authors dedicated a whole paragraph (lines 363 − 374) to explain the chemistry, and the complexity can also be seen in Table 1. While ESI is a universal technique and is able to ionize almost all molecules one would encounter in SOA, it will be difficult (or, at the very least, tedious) to work backwards and deduce the original molecules from the large set of ion formulas observed. The second problem is that ESI is not a quantitative technique, especially with direct infusion shown here without prior separation. ESI suffers from matrix effects, and it is difficult to use surrogate standards for quantification. Perhaps the authors should point out to readers that more quantitative ionization techniques should be used to fully exploit the usefulness of the 2D framework.

**[Responses]** We acknowledge the major ESI disadvantage in terms of quantification of analytes, and as suggested, we have added corresponding discussions to clarify the capability and proper applications of the ESI scheme, also given below. We would like to note that we use ESI due to its great compatibility with the high-voltage inlet of the drift tube as it can handle voltages higher than 10 kV so that the ESI emitter can be directly placed at the entrance of the drift tube dissolvation region. Also note that developing alternative ionization schemes that are compatible with the IMS drift tube is actually one of our main research focuses now.

 **[Changes]** Line 364-367: For species investigated in this study, their integral molecular structures are maintained during electrospray ionization. Quantification of these species requires

prior chromatographic separation to avoid matrix suppression on the analyte of interest (Zhang et al., 2016) or alternative ionization scheme that is compatible with the high-voltage IMS inlet and does not induce matrix effects.

[Responses] We agree with the reviewer that it is not quite straightforward to elucidate the molecular structure of the analytes based on their ionic formula from ESI. This is also why we spent a long paragraph to try to give readers more generalized information on the applicability of ESI scheme to most common functional groups in the atmosphere.

[Changes] Line 349-353: For species investigated in this study, their integral molecular structures are maintained during electrospray ionization. An exhibition of molecular formulas of ionic species is given in Table 1. Depending on the proton susceptibility of functional groups, amines, esters, and aromatic aldehydes are sensitive to the ESI(+) mode, whereas carboxylic acids and organic sulfates yield high signal-to-noise ratios in the ESI(−) spectra.

4. Resolving isomeric structures: leucine and isoleucine are biological molecules and there are many other techniques that are capable for resolving those compounds. I find the use of leucine and isoleucine to demonstrate the capability of isomer separation to be not too effective. Perhaps the authors can consider showing the capability of the IMS to separate molecules of atmospheric interest?

[Responses] We agree that using species of atmospheric interest to demonstrate the IMS capability of isomer separation would be more appropriate for the scope of this study, although chemical standards for most isomeric structures that have been proposed in the atmosphere are commercially unavailable. Nevertheless, our recent study (Krechmer et al., AMT, 2016) has demonstrated the mobility separation of IEPOX derived sulfate isomers that are produced from the reactive uptake of synthesized *trans/cis*-IEPOX onto the acidified ammonium sulfate seeds in chamber experiments. Figure 4 in Krechmer et al. (2016) shows three isomers with proposed chemical structures and their drift times.

In the revised manuscript, we have added the following sentences to point to the IMS data for the atmospheric relevant isomers. We think the leucine data carry merits in a way that isomer separation in both ESI positive (+) and negative (−) modes can be demonstrated due to the presence of both amine and carboxyl functional groups. Moreover, we show that the $\Omega_{N_2}$ values for the same isomer are different in these two ESI modes, depending on the functional group ($NH_2$ vs. $COOH$) that carries the charge. This is also consistent with the predictions by the trajectory method.

**[Changes]** Line 407-411: Here we demonstrate the separation of isomers on the $\Omega_{N_2} - m/z$ space using the mixture of L-leucine and D-isoleucine as an illustration, as they can be directly ionized by electrospray in both positive and negative modes due to the presence of amino and carboxyl groups. We refer the reader to Krechmer et al. (2016) for the mobility separation of atmospheric relevant isomeric species.

Minor:

Figure 2: it seems to me that citric acid and the tricarboxylic acid are not quite the same (differs by an –OH group, the other homologue series differs by one or more –CH2- groups) so the trend line should not be drawn the same way as in the other series.

**[Responses]** Corrected.

Figure 4: I don't understand the fragmentation pattern in the amine (di-tert-butyl pyridine). Why is there a loss of –CH3 group? (It will form an unstable ion with an unpaired electron.)

**[Responses]** As shown in the figure below, the unpaired electron on the carbon atom resulted from the neutral loss –CH$_3$ group can be stabilized by conjugation with the pyridine ring. The Time-of-Flight (TOF) extraction period is 69 µs. Thus we think it is a reasonable assumption that such a radical could survive within the order of micro-seconds before being detected.

---

## Author Comment (AC2) · 21 Sep 2016

**Response to Reviewer #1**

In this manuscript, the authors present a new framework with which to identify and characterize atmospherically relevant organic compounds based on a combination of ion mobility and molecular mass. Though a wide variety of such two-dimensional frameworks have been employed to parameterize and simplify descriptions of atmospheric mixtures, the technique proposed by the authors is unique and valuable in its ability to characterize compounds based on structural features and to separate isomeric species. The authors have done an excellent and detailed job of exploring intrinsic spatial relationships in this parameter space. Continued application of the tools and techniques described in this manuscript will likely provide more molecular and chemical information than has previously been available.

*We thank reviewer #1 for the constructive and insightful comments. Our point-by-point responses can be found below, with reviewer comments in **black**, our responses in **blue**, alongside the relevant revisions to the manuscript in **red**.*

General comments:

While this manuscript builds a strong foundation for the application of these techniques to atmospheric samples, no attempt is made to apply these techniques to complex mixtures of unknowns. The title, abstract, and some portions of the introduction should be re-framed to highlight what is actually in this manuscript and focus less on what the authors hope to do with these tools in the future. It is implied or, in the case of the title stated explicitly, that this paper is about the "Characterization of Atmospheric Organic Aerosol." Given the home institutions of the authors, I have no doubt this is the goal and am excited to see it applied to ambient mixtures. However, without more detailed attempts to apply this technique to atmospheric mixtures or at least detailed discussion, the language and title of this manuscript should be changed to focus more on "atmospherically relevant organic compounds," or "highly oxidized small organic compounds," or "characterizing functionality of organic compounds." The detailed description of the framework also needs some added clarity – see detailed comments below.

**[Responses]** The reviewer has raised a good point. As we have not applied this framework to deconvolute the complexity of atmospheric aerosol mixture, the focus of 'characterizing organic aerosols' becomes irrelevant to the scope of the current study. On the other hand, this framework is not limited to the condensed phase speciation but can be applied to chemicals in the environment at all forms, with the use of appropriate ionization schemes. In fact, developing

alternative ionization schemes that are suitable for gas-phase measurement and also compatible with IMS drift tube is one of our current research focuses. As suggested, we have changed the language of the entire text to focus on the atmospherically relevant organic compounds, as all the functional groups characterized in the study have been identified as major components in the atmosphere.

We would also like to note that progresses have been made in applying this framework to ambient organic aerosol mixtures. In fact, the application of IMS-MS to a rather simple SOA system that is generated from the reactive uptake of IEPOX onto acidified ammonium sulfate seed particles has been demonstrated by our recent study (Krechmer et al. AMT, 2016). Figure 5c in Krechmer et al. (2016) shows the distribution of IEPOX derived organosulfate, along with its dimers and trimers, on the 2-D space. A trend line for the major IEPOX monomers, dimers, and trimmers is clearly visible. The characteristic fragment upon CID, i.e., sulfate (m/z 97), adds further confirmation on the chemical identities of the IEPOX derived products in the particle phase.

[Changes] Title: A Novel Framework for Molecular Characterization of Atmospherically Relevant Organic Compounds Based on Collision Cross Section and Mass-to-Charge Ratio

Abstract: … "A new metric is introduced for representing the molecular signature of atmospherically relevant organic compounds." … "Reactions involving changes in functionalization and fragmentation can be represented by the directionalities along or across these trend lines, thus allowing for the interpretation of atmospheric transformation mechanisms of organic species." …

Introduction: Organic species in the atmosphere — their chemical transformation, mass transport, and phase transitions — are essential for the interaction and coevolution of life and climate (Pöschl and Shiraiwa, 2015). Organic species are released into the atmosphere through biogenic processes and anthropogenic activities.  Once in the atmosphere, organic species actively evolve via multiphase chemistry and gas-particle phase conversion. The complexity and dynamic behaviors of organic species have prevented our capability to accurately predict their levels, temporal and spatial variability, and oxidation dynamics associated with the formation and evolution of organic aerosols in the atmosphere.

Minor comments:

Line 40: "scatter plots" is not a verb

**[Responses]** We refer to the definition of van Krevelen diagram in Heald et al., GRL, (2010): "*The Van Krevelen diagram was developed to illustrate how elemental composition changes during coal formation [Van Krevelen, 1950]. The diagram cross plots the hydrogen to carbon atomic ratio (H:C) and the oxygen to carbon atomic ratio (O:C).*"

Line 64: Re-word, perhaps use "and subsequent interactions" in place of "as well as"

**[Responses]** Revised as suggested.

Line 169: This phrase is awkwardly broken up and should be re-worded: "the instrument standard (the reduced mobility of such a standard is not affected by contaminants in the buffer gas) is needed"

**[Responses]** Agreed and revised as follows.

**[Changes]** Line 168-173: In view of these uncertainties, the instrument standard is needed to provide an accurate constraint on the instrumental parameters, such as voltage, drift length, pressure, and temperature.

$$K_0 \times t_d = \frac{L_d^2}{V_d} \frac{P}{1013.25} \frac{273.15}{T} = C_i \tag{5}$$

Tetraethyl ammonium chloride (TEA) is used here as the instrument standard, as its reduced mobility is not affected by contaminants in the buffer gas.

Lines 184-200 describe the apparent crux of this framework, but some points could be made clearer. In particular, explicitly relating the measured parameters to the calculated parameters would be very helpful. For instance, in Eq. 6, it would be useful to re-frame in terms of t_d since that is what is actually being measured, instead of K_0 and v_d. What is the functional form of this relationship, considering all of the terms in the equation? Is it collision cross section generally linear with drift time?

**[Responses]** It is the arrival time ($t_a$) that is ultimately measured under certain potential gradients applied to the drift tube ($V_d$). The arrival time ($t_a$) is the sum of the time that the ion spent in the drift tube ($t_d$) and the time for the ion from the exit of the drift tube to the detector ($t_0$). Knowing $t_a$ and $V_d$, the mobility (K) can be derived through linear regression of Equation (3). The reviewer is correct that the collision cross section is a linear function of the drift time. But since the drift time is not a directly measured quantity, one step of linear regression is

required to derive $t_d$ or $K$. In view of the potential complexity in the calculations, we have attached Matlab codes in calculating the collision cross section in the Supplementary Material. In addition, the physical meaning of all parameters in the manuscript is attached in the Appendix.

[Changes] Line 198: Matlab codes for calculating $\Omega_{N_2}$ are given in the Supplement.

Line 187: Is z=1 assumed for all ions? In contrast to other ionization techniques, ESI can under some conditions yield a distribution of charges – is this an issue and to what extent would it change the results?

[Responses] Yes, we have carefully verified that for all the standards characterized in this study, no multiple charged ions are observed in the mass spectra. Electrospray ionization is conducive to the formation of singly charged small molecules (those we care most about) but is also well known for producing multiply charged species of larger molecules. Fortunately, software available with all electrospray mass spectrometers facilitates the molecular weight calculations necessary to determine the mass of the multiply charged species. In terms of distribution of singly charged vs. multiply charged ions on the space, previous studies (e.g. *Pringle et al., Int. J. Mass, 2007*) have demonstrated that they are well resolved because they lie along trend lines with different slopes.

Line 190: Is thermal velocity calculated as a function of molecular mass?

[Responses] Yes, it is a function of molecular mass and temperature. Matlab codes for calculating this quantity is attached in the Supplementary Material.

Line 193: How are the mass fractions calculated? How might this work for a mixture of unknowns with poorly defined sensitivities?

[Responses] We define $\widehat{m}$ and $\widehat{M}$ are molecular mass fractions of the ion and the buffer gas molecule ($N_2$), respectively:

$$\widehat{m} = \frac{m}{m + M} \qquad \widehat{M} = \frac{M}{m + M}$$

where m and M are the molecular masses for the ion and the buffer gas molecule ($N_2$), respectively. Note that the word 'mass' here is not the actual concentration for unknowns in a mixture, but the molecular weight for unknowns, which are derived from the measured mass-to-charge ratio. We apologize for the confusion and have clarified this in the manuscript.

Line 214: Define or clarify "(12,4) potential"

[Responses] We define the (12,4) potential along with Equation (9).

[Changes] Line 214-215: The potential during interaction includes a long-range polarization term and a short-range repulsion term (Mason et al., 1972)."

Line 245-249: The core model, consisting of a (12-4) central potential displaced from the origin, is used to represent interactions of polyatomic ions with $N_2$ molecules (Mason et al., 1972). The (12-4) central potential includes a repulsive $r^{-12}$ term, which describes the Pauli repulsion at short ranges due to overlapping electron orbitals, as well as an attractive $r^{-4}$ term, which describes attractions at long ranges due to ion induced dipole:

Sections 3.1 and 3.2 could perhaps be switched, as they are discussed in Section 3 and in Section 4 in the opposite order.

[Responses] We would like to maintain the original order for Section 3.1 and 3.2, as the core model is basically a simplified version of the trajectory method. We feel that Equation (8) should be introduced first to give the readers a complete overview on the description of ion-neutral interaction potentials before moving on to Equation (9), which simplifies the ion structure as a sphere and ignores the $r^{-6}$ term (attraction at long ranges due to van der Waals force) as it becomes inappreciable compared with attraction due to ion induced dipole.

Line 301: It is not clear to me that "functionalization and fragmentation can be represented by an intrinsic directionality" as claimed by the authors. As the authors note, the connected markers shown in Figure 2 represent addition of non-functionalized carbon atoms, but this is not the form that atmospheric functionalization takes. Instead, addition of carbon moves up and to the right, but addition of functional groups appears to move down. What would a vector of functionalization or fragmentation look like in this space? This question is particularly important if the authors intend to keep their focus on using this approach to characterize complex mixtures.

[Responses] As the reviewer suggested, we have added a vector of functionalization and fragmentation on the 2-D space in Figure 3. Here we define the functionalization as the addition of oxygen-containing functional groups and fragmentation as the cleavage of C-C bonds. As we show that molecules with lower collision cross section are generally much denser and more functionalized, the trajectory for functionalization on the space would be mostly downward, with a shift to the right hand side considering the increase of molecular mass. The trajectory for

fragmentation can be illustrated by the trendline for $C_8$-$C_{18}$ mono-carboxylic acid shown in Figure 2. As fragmentation only involves changes in carbon number, the corresponding trajectory would more or less follow this trendline.

[Changes] Line 329-342: 4.3 Trajectories for Atmospheric Transformation Processes

Functionalization (the addition of oxygen-containing functional groups) and fragmentation (the oxidative cleavage of C–C bonds) are key processes during atmospheric transformation of organics. Reactions involving changes in functionalization and fragmentation can be represented by directionalities on the $\Omega_{N_2} - m/z$ space, as illustrated by the distribution pattern of carboxylic acids in Figure 3. Addition of one carbon atom always leads to an increase in mass and collision cross section, with a generic slope of approximately 5 $\mathring{A}^2$/Th. Although the addition of one oxygen atom in the form of a carbonyl group results in a similar increase in the molecular mass, it leads to a shallower slope compared with that from expanding the carbon chain. Addition of carboxylic or hydroxyl groups does not necessarily lead to an increase in the collision cross section, as the formation of the intramolecular hydrogen bonding ($O - H \cdots O^-$) could result in a more compact conformation of the molecule. In general, fragmentation moves materials to the bottom left and functionalization to the right on the space.

Line 345: While the trend lines described in the Section 4.2 provide an important characterization of this framework, and a useful test of the core model, it's not completely clear they would be particularly help in identifying unknown species, as implied in this sentence. As demonstrated by Figure S5, there is substantial overlap between the regions of all of the trend lines – if one were handed an unknown, its location in this space alone would not provide much information on its family. These lines are perhaps useful for identifying family of species, and demonstrate the utility of the core model for helping to understand its location in the space, but as written this sentence is a bit of an overstatement without some explanation or support. Section 4.3, on the other hand, does indeed seem very promising for identifying unknowns…

[Responses] The following steps illustrate how we could potentially identify unknowns in a mixture:

1). We first try both positive (+) and negative (−) ESI modes and identify the ideal mode in which the sample would yield intensive signals. As we have demonstrated in Section 4.4, amines, aldehydes, and esters can only be detected in positive (+) mode, while carboxylic acids and sulfate yield high signal to noise ratios in negative (−) mode. This gives us primary information on the functionalities an unknown might contain, and more importantly, avoids

overlaps that would have occurred if one mixed data from two modes together, as shown in Figure S5.

2). We then measure the collision cross section ($\Omega_{N_2}$) of the sample and map its location on the $\Omega_{N_2} - m/z$ space. Based on the predicted trend lines that are constrained by chemical standards, we could possibly identify the chemical families to which the unknowns belong.

3). If there are overlaps of species even in one ESI mode, we need to use the collision induced dissociation (CID) function to identify the characteristic fragments for the overlapping species. For example, *mono*-carboxylic acids yield $CO_2$ (44 amu), di-carboxylic acids yield both $CO_2$ and $H_2O$ (18 amu), sulfates yield $HSO_3^-$ (97 amu).

In summary, a combined knowledge on the mass to charge ratio of the unknowns, the location of the unknowns on the 2-D space, as well as the fragmentation patterns of unknowns, could provide information on the chemical classes to which the unknowns belong as well as the functional groups the unknowns contain.

**[Changes]** We have rewritten Section 4.1 and 4.2, by clarifying the application of 2-D space to the identification of chemical classes not the molecular structures of unknowns.

Line 318-328: The demonstrated $\Omega_{N_2} - m/z$ trend lines provide a useful tool for categorization of structurally related compounds. Mapping out the locations and distribution patterns for various functionalities on the 2-D space would therefore facilitate classification of chemical classes for unknown compounds. It is likely that trend lines extracted from a complex organic mixture overlap and, as a result, the distribution pattern of unknowns on the space alone would not provide sufficient information on their molecular identities. In this case, the fragmentation pattern of unknowns upon collision induced dissociation (CID) needs to be explored for the functionality identification, as discussed in detail in Section 4.4. As it is highly unlikely that two distinct molecules will produce identical IMS, MS, as well as CID-based MS spectra, the 2-D framework therefore virtually ensures reliable identification of species of atmospheric interest.

Line 386: A dominant peak at 108 is mentioned but not shown in Figure 4.

**[Responses]** We have added the measured drift time for ion at *m/z* 108 in Figure 4, also given below:

[Figure]

Line 398: The there is no O-O bond in dioctyl phathalate. Do the authors mean the carbonyl-oxygen bond? Interestingly (and relatedly), this 149 peak is the dominant peak in EI spectra.

**[Responses]** Yes, it should be carbonyl-oxygen bond. We have corrected this in the manuscript. Yes, the *m/z* 149 peak is also dominant in the EI spectra of dioctyl phthalate, as shown in the Aerodyne Mass Spectrometer (AMS) collected mass spectra below. We tentatively propose the following structure for the *m/z* 149 ion, and perhaps it is also produced from the fragmentation during EI.

[Figure]

*Proposed structure for m/z 149*

*Data source: Aiken et. al. Anal. Chem., 2007.*